# Administration of *Momordica charantia* Enhances the Neuroprotection and Reduces the Side Effects of LiCl in the Treatment of Alzheimer’s Disease

**DOI:** 10.3390/nu10121888

**Published:** 2018-12-03

**Authors:** Hei-Jen Huang, Shu-Ling Chen, Yen-Ting Chang, Jong-Ho Chyuan, Hsiu Mei Hsieh-Li

**Affiliations:** 1Department of Nursing, Mackay Junior College of Medicine, Nursing and Management, Taipei 11260, Taiwan; camera0207@yahoo.com.tw; 2Department of Life Science, National Taiwan Normal University, Taipei 11677, Taiwan; minnie13689@gmail.com (S.-L.C.); siphinie@gmail.com (Y.-T.C.); 3Hualien District Agriculture Research and Extension Station, Hualien 97365, Taiwan; jonghoc@hdares.gov.tw

**Keywords:** lithium chloride, *Momordica charantia*, streptozotocin, neuroprotection, survival, Alzheimer’s disease

## Abstract

Recently, the use of natural food supplements to reduce the side effects of chemical compounds used for the treatment of various diseases has become popular. Lithium chloride (LiCl) has some protective effects in neurological diseases, including Alzheimer’s disease (AD). However, its toxic effects on various systems and some relevant interactions with other drugs limit its broader use in clinical practice. In this study, we investigated the in vitro and in vivo pharmacological functions of LiCl combined with *Momordica charantia* (MC) in the treatment of AD. The in vitro results show that the order of the neuroprotective effect is MC5, MC3, MC2, and MC5523 under hyperglycemia or tau hyperphosphorylation. Therefore, MC5523 (80 mg/kg; oral gavage) and/or LiCl (141.3 mg/kg; intraperitoneal injection) were applied to ovariectomized (OVX) 3×Tg-AD female and C57BL/6J (B6) male mice that received intracerebroventricular injections of streptozotocin (icv-STZ, 3 mg/kg) for 28 days. We found that the combined treatment not only increased the survival rate by reducing hepatotoxicity but also increased neuroprotection associated with anti-gliosis in the icv-STZ OVX 3×Tg-AD mice. Furthermore, the cotreatment with MC5523 and LiCl prevented memory deficits associated with reduced neuronal loss, gliosis, oligomeric Aβ level, and tau hyperphosphorylation and increased the expression levels of synaptic-related protein and pS9-GSK3β (inactive form) in the icv-STZ B6 mice. Therefore, MC5523 combined with LiCl could be a potential strategy for the treatment of AD.

## 1. Introduction

Alzheimer’s disease (AD) is the most common multi-factorial neurodegenerative disorder worldwide devastatingly affecting the aged population. AD is characterized by the loss of cholinergic neurons, amyloid-beta peptide (Aβ) plaques and neurofibrillary tangles, and several hypotheses have been proposed to explain the pathogenesis of AD, including cholinergic, amyloid, and metal ions [1,2,3]. However, current therapeutic drugs for AD, such as acetylcholinesterase inhibitors and NMDA antagonists, merely provide symptomatic treatment and do not target the underlying cause of the disease [4]. This frustrating situation prompts the reconsideration of therapeutic strategies against AD. Evidence shows that individuals with type 2 diabetes (T2D) have a nearly two-fold higher risk of developing AD than non-diabetic individuals [5]. Insulin resistance in T2D induces inflammatory responses, which are further intensified under hyperglycemia and promote long-term diabetic complications [6]. In addition, the overexpression of inflammatory mediators, such as TNF-α, IL-6 and IL-1β, may trigger signaling cascades in neurons leading to the activation of protein kinases, including GSK3β and CDK5, resulting in the hyperphosphorylation and self-aggregation of tau protein into neurotoxic oligomeric species [7]. Therefore, AD is often termed type 3 diabetes due to its pathophysiological similarities to T2D. Evidence also demonstrates that icv-STZ triggers central insulin resistance as a sporadic Alzheimer-like (sAD) pathology [8,9] and exacerbates the impairment of memory deficits in 3×Tg-AD mice [10]. In this study, icv-STZ was used to establish an sAD animal model and accelerate the pathological progression in 3×Tg-AD mice. 

GSK3β kinase serves as a powerful therapeutic target in diabetes and AD [11]. The hyperphosphorylation of tau protein is caused by an imbalance between the phosphorylation and dephosphorylation of tau by overactivated GSK3β [12]. A previous study showed that toxic Aβ increased the activity of GSK3β, which induced tau protein phosphorylation [13]. Therefore, GSK3β plays an important role linking Aβ toxicity to tau pathology [12]. Recently, small molecule inhibitors with GSK3β activity, such as Anthraquinones, Benzothiazoles, and Phenothiazines, were demonstrated to dissolve preformed tau aggregates [14]. However, anti-aggregation usually leads to the generation of small oligomers, which are currently hypothesized to be even more harmful [15,16]. Lithium chloride (LiCl) is a conventional GSK3β inhibitor that reduces tau phosphorylation in animal models [17] and prevents cognitive decline in the treatment of bipolar disorder [18]. Additional evidence suggests that LiCl has therapeutic efficacy in AD models [19,20] and mood disorders [21,22], followed by severe side effects, such as neurological toxicity [23]. Moreover, LiCl is highly toxic at conventional doses, especially among elderly individuals, which severely limits its application in the treatment of AD [24]. Combination therapies have been suggested to diminish the toxicity of LiCl and increase the potential of LiCl in the treatment of AD [23]. Evidence further indicates that combination therapeutic strategies offer both efficacious and safe treatment in AD patients [25,26]. Therefore, this study focused on the administration of natural food supplementation to reduce the toxicity of LiCl and delay disease onset in AD patients. 

Wild bitter gourd (*Momordica charantia*; MC) is a common vegetable in Asia that is used in traditional medicine to treat various diseases, including inflammation [27], diabetes [28], and cancer [29]. A previous study has shown that MC has potent neuroprotective activity against neurological deficits in diabetic mice [30]. Both inflammation and diabetes are risk factors associated with AD [31]. In addition, several studies have shown that MC has low toxicity following oral intake [32,33]. Thus, a therapeutic strategy comprising MC combined with LiCl is a reasonable and potential option for the treatment of AD. The ethanol extracts of wild bitter gourd cultivars (Hualien Nos. 3 and 4) have been shown to have beneficial effects against alcoholic fatty liver disease by attenuating oxidative stress and the inflammatory response [34]. Moreover, Hualien No. 4 has been shown to induce antidiabetic activity in experimental settings ranging from in vitro to humans [35,36]. In this study, we tested 4 different wild bitter gourd cultivars (Hualien Nos. 2, 3, 5 and 5523) and found that the neuroprotective effects of MC5 were superior to those of the other MC cultivars in mouse primary culture under hyperglycemia. However, the combinational effects of LiCl with MC5523 were better than those of MC5 in mouse primary hippocampal neurons following tau hyperphosphorylation. Furthermore, the administration of MC5523 plus LiCl not only contributed to a delayed disease onset and neuroprotection in the icv-STZ B6 mice but also increased the survival rate and reduced hepatotoxicity in the icv-STZ ovariectomized (OVX) 3×Tg-AD mice. Therefore, MC5523 combined with LiCl could be a potential therapeutic strategy for AD. 

## 2. Materials and Methods

### 2.1. Animals

C57BL/6J and 3×Tg-AD (harboring PS1_M146V_, APP_swe_ and Tau_P30IL_ transgenes) mice were purchased from the National Laboratory Animal Center (NLAC; Taipei, Taiwan) and the Jackson Laboratory (004807), respectively. An acute intracerebroventricular injection of streptozotocin (icv-STZ, 3 mg/kg; Sigma, St. Louis, MO, USA) was applied to the C57BL/6J mice (8 weeks old) to establish a sporadic AD mouse model [37]. The same dose of icv-STZ was also applied to OVX 3×Tg-AD female mice (12 months old) to exacerbate the pathological progression as previously reported [10]. The mice were housed at 20–25 °C with 60% relative humidity under a 12-h light/dark cycle, and food and water were available ad labium. All experiments were performed during the light phase between 7:00 a.m. and 7:00 p.m. The mice were deeply anesthetized with avertin (0.4 g/kg of body weight) and then sacrificed for the pathological analyses after a series of behavioral tasks. The animal experiments were conducted in accordance with the Institutional Animal Care and Use Committee (IACUC) of National Taiwan Normal University, Taipei, Taiwan (Permit Number: 103004). All efforts were made to minimize suffering.

### 2.2. Preparation of Wild Bitter Gourd Powder

All four MC strains (Hualien Nos. 2, 3, 5, and 5523) were obtained from the Hualien District Agricultural Research and Extension Station, Council of Agriculture, Executive Yuan, Taiwan. The characteristics of MC5523 (a novel wild bitter gourd cultivar) and MC5 are shown in Appendix A. Whole fruits from each cultivar were cut into small pieces. The pieces were collected, freeze-dried and finely ground. The freeze-dried powders were dissolved in saline and used in the animal studies via oral gavage (o.g.).

### 2.3. Animal Experimental Design

After 6 days of o.g. adaptation, the B6 and OVX 3×Tg-AD mice were randomly divided into the following 5 groups (12–16 mice/group): (i) saline (icv)/saline (o.g.)/saline (intraperitoneal injection; i.p.); (ii) STZ (icv)/saline (o.g.)/saline (i.p.); (iii) STZ (icv)/MC5523 (o.g.)/saline (i.p.); (iv) STZ (icv)/saline (o.g.)/LiCl (i.p.); and (v) STZ (icv)/MC5523 (o.g.)/LiCl (i.p.). Considering the effective dose in mouse primary hippocampal neuronal culture treated with MC5523 (10 mg/ml) and LiCl (5 mM), the fraction of drug absorbance trough oral gavage (o.g.) (approximately 15%) or intraperitoneal injection (i.p.) (approximately 60%), the releasing fraction from the protein association in the plasma (approximately 10%), and the bodyweight of the mice, MC5523 (80 mg/kg; o.g.) and LiCl (141.3 mg/kg; i.p.; Sigma) were applied to the mice. At 4 h after the MC5523 and/or LiCl administration, the mice were anesthetized with avertin (0.4 g/kg of body weight; Sigma) and received a single injection (icv) of 3 µL of STZ (3 mg/kg) into the left lateral ventricle. The bregma coordinates used for the injection were −1.0 mm lateral, −0.3 mm posterior, and −2.5 mm below. The control mice received an equal volume of normal saline. MC5523 (80 mg/kg; o.g., daily) and LiCl (141.3 mg/kg; i.p., daily) were applied to the mice for 28 days (days 7–34). The Y maze and Morris water maze (MWM) tests were conducted on day 26 and days 28–34, respectively. Finally, the mice were sacrificed for the western blot and immunohistochemical analyses on day 35 (Figure 1A).

### 2.4. Y Maze

The Y maze testing was performed as previously described [38] using an apparatus with three equal arms (35 cm long × 5 cm wide × 15 cm high) made of white acrylic. After the handling adaptation, the mice were individually placed at the center of the maze and allowed to explore for 8 min (*n* = 12–15 per group). The mice that stayed at the center of the maze during the experiment were excluded from the experiment. The series of arm entries were recorded visually, and the alternation percentage was calculated. The total number of entries (N) and the number of “correct” triplets (M, consecutive choices of each of the three arms without re-entries) were evaluated. The alternation rate (R) was computed according to the formula R (%) = [M/(N − 2)] × 100%.

### 2.5. Morris Water Maze (MWM)

Spatial learning and memory were evaluated using a conventional MWM as previously described [39,40,41]. During the MWM training, an escape platform (10 cm in diameter) made of white plastic was submerged 1.0 cm below the water level. The swim path of each mouse during each trial was recorded by a video camera connected to a video tracking system (Noldus, Wageningen, Netherlands). On the day prior to the spatial training, all mice underwent pretraining to assess their swimming ability and acclimatize the mice to the pool (*n* = 12–15 per group). The mice that floated in the pool during the pretraining stage were excluded from the experiment. A 4-day training session consisting of four 60-s training trials (inter-trial interval: 20–30 min) per day was conducted with a hidden platform placed at the same location in the pool (northeast quadrant). The mice that failed to locate the platform within 60 s were placed on the platform for 20 s during the training period. The escape latency time to reach the platform was recorded in each trial. Three probe trials were performed 48 h after the final training trial. During the probe trial, the mice were allowed to swim for 60 s after the platform was removed from the pool. The platform-crossing frequencies were recorded to evaluate the changes in long-term spatial memory in each group.

### 2.6. Immunohistochemistry

After the MWM test (day 35), the mice (*n* = 3–5 per group) were anesthetized (avertin; 0.4 g/kg) and transcardially perfused with 4% paraformaldehyde in phosphate-buffered saline (PBS). The mouse brains were removed, post-fixed with 4% paraformaldehyde for 4 h, cryo-protected with 10% sucrose for 1 h, followed by 20% sucrose for 2 h, and then placed in 30% sucrose in PBS for 2 days. Then, the samples were subjected to continuous serial cryostat sectioning at 30 µm by a microtome (CMS3050S, Leica Microsystems, Nussloch, Germany). The specific primary antibodies used are listed in Table 1. Free-floating sections were used for the immunohistochemistry staining as previously described [39,40]. Nonspecific epitopes were blocked by incubation with 5% normal goat or rat serum and 0.1% Triton X-100 in PBS for 1 h. Then, the sections were incubated with primary antibodies overnight at room temperature, secondary antibodies (1:200 dilution in blocking solution, Vector Laboratories, Burlingame, CA, USA) for 1 h, and then an avidin-biotin complex for 1 h at room temperature. The reaction was developed using a 3,3′ diaminobenzidine (DAB) kit (Vector). All sections were mounted on coated slides and cover-slipped for light microscopy. Positive neuron staining in a specific area was first selected as a standard signal, and then, the numbers of neurons stained positive for the above antibodies were counted using digital image analysis software (Image-Pro Plus, Media Cybernetics, Rockville, MD, USA). The pixel counts were calculated as the average of three adjacent sections per animal.

### 2.7. Preparation of Liver Samples

Liver tissues were isolated from all groups and fixed by immersion in 4% paraformaldehyde solution at room temperature for 24 h. After fixation, blocks of liver tissues were embedded in paraffin for a routine histological examination. The paraffin-embedded tissue was cut into 5-μm-thick sections and stained with H&E prior to examination by light microscopy. 

### 2.8. Western Blot Analysis

The protein was extracted from the hippocampus of the mice (*n* = 3–5 per group). The amount of protein was determined using a bicinchoninic acid (BCA) protein assay kit (Pierce). The protein (50 μg) was separated by sodium dodecyl sulphate-polyacrylamide gel electrophoresis (SDS-PAGE) and transferred to polyvinylidene difluoride (PVDF) membranes. The blots were probed with various primary antibodies as listed in Table 1. The same blot was probed for the housekeeping protein β-actin, which served as a loading control. Secondary antibodies, including anti-rabbit IgG HRP-conjugated antibody (1:10,000; Amersham Pharmacia Biotech; Piscataway, NJ, USA) and anti-mouse IgG HRP-conjugated antibody (1:10,000; Amersham Pharmacia Biotech; NJ, USA), were used. The specific antibody-antigen complexes were detected by an enhanced chemiluminescence detection system (Amersham Pharmacia Biotech; NJ, USA). The quantification was performed using an LAS-4000 chemiluminescence detection system (Fujifilm; Tokyo, Japan), and the target protein density was normalized to that of the internal control β-actin.

### 2.9. Data Analysis

All data are expressed as the mean ± standard error of the mean. In the comparison of the 4-day learning curve (training period), the presence or absence of a simple main effect was determined using a one-way analysis of variance (ANOVA). Subsequently, for cases in which a simple main effect was observed to be significant, the points displaying significant differences were identified using a least significant difference (LSD) post hoc test. Kaplan–Meier survival curves of the mice in the different treatment groups were analyzed by a log rank survival test. The statistical analyses were performed using SPSS 15.0 software (SPSS Inc., Chicago, IL, USA). A *p*-value  < 0.05 was considered significant.

## 3. Results

### 3.1. Administration of MC5523 Enhances Neuroprotection and Reduces Hepatotoxicity Induced by LiCl in icv-STZ OVX 3×Tg-AD Mice

Our in vitro results showed that the order of the neuromorphological protective effect against damage induced by hyperglycemia in mouse primary culture is MC5, MC3, MC2, and MC5523 (Appendix A). However, the combinational effect of LiCL and MC5523 is better than that of LiCL and MC5 in promoting the growth of both neurite length and branching (Appendix A). Therefore, the treatment with LiCl combined with MC5523 was applied in vivo. To evaluate the therapeutic benefits of MC5523 combined with LiCl on the survival rate, OVX 3×Tg-AD mice were pretreated with MC5523, LiCl, MC5523 plus LiCl or saline (vehicle) 4 h before the icv-STZ treatment. During the experimental period, the combination of MC5523 and LiCl greatly increased the mouse survival rate against the lethal threat of LiCl in the STZ-treated group (*p* < 0.05; Figure 1B). LiCl has been reported to increase apoptosis in liver tissue [38]; therefore, we examined whether the survival benefit of MC5523 was associated with the hepatotoxicity induced by LiCl (i.p.) in the icv-STZ OVX 3×Tg-AD mice. We found that LiCl administered by an intraperitoneal injection induced hepatotoxicity and that the combination of LiCl and MC5523 ameliorated hepatotoxicity in the icv-STZ OVX 3×Tg-AD mice (Figure 1C). Therefore, the administration of MC5523 increased the survival rate by reducing the hepatotoxicity induced by LiCl in the icv-STZ OVX 3×Tg-AD mice.

### 3.2. Cotreatment with MC5523 and LiCl Greatly Prevents Neuronal Loss in icv-STZ OVX 3×Tg-AD Mice

Compared with the icv-saline treatment, the icv-STZ treatment significantly induced neuronal loss in the hippocampal CA1 subregion in the OVX 3×Tg-AD mice (*p* < 0.001; Figure 2A,B). The administration of either MC5523 or LiCl alone or in combination reduced this neuronal loss; MC5523 plus LiCl had a better effect than either MC5523 or LiCl alone (*p* < 0.01; Figure 2A,B). Compared with the saline/saline/saline group, we also observed a significant loss of serotonergic neurons in the STZ/saline/saline group (*p* < 0.001; Figure 2A,C). Moreover, the LiCl treatment alone exaggerated this loss of serotonergic neurons (*p* < 0.01; Figure 2A,C). However, both MC5523 alone and MC5523 with LiCl attenuated the loss of serotonergic neurons induced by the STZ or LiCl treatment (*p* < 0.001; Figure 2A,C), suggesting that the combined effect of MC5523 and LiCl is better than that of MC5523 alone (*p* < 0.001; Figure 2A,C). In the locus coeruleus (LC) region, only the cotreatment with MC5523 and LiCl, but not the treatment with MC5523 or LiCl alone, significantly prevented the loss of noradrenergic neurons induced by STZ (*p* < 0.001; Figure 2A,D). Thus, cotreatment with MC5523 and LiCl largely prevented the neuronal loss observed in the hippocampal CA1, Raphe, and LC regions of the icv-STZ OVX 3×Tg-AD mice. 

### 3.3. Cotreatment with MC5523 and LiCl Greatly Reduces Gliosis in icv-STZ OVX 3×Tg-AD Mice

The aberrant activation of microglia and astrocytes plays an important role in chronic neuroinflammation and exacerbates disease progression [42]. We found that the number of activated astrocytes and microglia was significantly increased in the STZ/saline/saline group compared with that in the saline/saline/saline group (*p* < 0.001; Figure 3A–C). The number of astrocytes was significantly reduced after the treatment with MC5523 alone and in combination with LiCl (*p* < 0.001; Figure 3A,B); additionally, the cotreatment with MC5523 and LiCl had a better effect than the treatment with MC5523 alone (*p* < 0.05; Figure 3A,B). Furthermore, the mice treated with MC5523, LiCl, and MC5523 plus LiCl showed significantly fewer activated microglia than the saline-treated mice under the icv-STZ condition (*p* < 0.001; Figure 3A,C). In addition, the cotreatment with MC5523 and LiCl significantly reduced the number of activated microglia compared with the treatment with MC5523 or LiCl alone (*p* < 0.01; Figure 3A,C). Thus, we suggest that cotreatment with MC5523 and LiCl greatly reduces gliosis in icv-STZ OVX 3×Tg-AD mice.

### 3.4. Combination of MC5523 and LiCl Shows More Beneficial Effects on Short-Term Memory in icv-STZ B6 Mice

To further confirm the beneficial effects of MC5523 plus LiCl, the icv-STZ male B6 mice were used as a sporadic AD model as previously reported [43]. In the Y maze task (performed on day 26), the mice were tasked with selecting a pathway in the Y-shaped track. The spontaneous alterations in arm entries in the icv-STZ mice were significantly lower than those in the icv-saline mice (*p* < 0.05; Figure 4A). Notably, the performance in spontaneous alterations was improved in the mice after the treatment with LiCl, MC5523, or LiCl plus MC5523 compared with that in the saline-treated icv-STZ mice (*p* < 0.05–0.001; Figure 4A). In particular, the treatment with MC5523 and LiCl greatly increased the spontaneous alterations compared with the treatment with LiCl or MC5523 alone (*p* < 0.01; Figure 4A). The number of arm entries did not substantially differ among all groups (*p* > 0.05; Figure 4B), indicating that spontaneous alterations in behavior are not caused by changed movements.

### 3.5. Combination of MC5523 and LiCl Improves Spatial Cognition in icv-STZ B6 Mice

The spatial cognitive effects of LiCl, MC5523, and LiCl plus MC5523 in the WMW test were elucidated and are shown in 4C–E . Initially, there were no differences in the swimming velocity among the groups (*p* > 0.05; Figure 4C), indicating that the spatial cognition measurements were obtained under normal motor function conditions. During the training period (days 29–32), we observed that as the training time extended, the escape latency of the mice in the saline/saline/saline (control) (F(3, 47) = 4.103; *p* < 0.05; Figure 4D) and STZ/LiCl/MC5523 (F(3, 31) = 4.091; *p* < 0.05; Figure 4D) groups showed a decreasing trend. The escape latency of mice in the STZ/saline/saline, STZ/LiCl/saline and STZ/MC5523/saline groups showed marked retardation from training day 3 to 4 (Figure 4D). To investigate the effect on long-term spatial memory, performance during the probe trial (day 34) was examined by analyzing the platform-crossing frequency. The mice in the STZ/saline/saline group showed significantly fewer platform crossings during the probe trials, which is indicative of memory impairment after the icv-STZ intervention (*p* < 0.05; Figure 4E). However, the platform-crossing frequency in the STZ/LiCl/MC5523 group was significantly higher than that in the STZ/saline/saline group (*p* < 0.05; Figure 4E). Furthermore, the platform-crossing frequency in the STZ/LiCl/MC5523 group was significantly greater than that in the STZ/LiCl/saline and STZ/MC5523/saline groups (*p* < 0.01; Figure 4E). These results from the MWM test suggest that cotreatment with MC5523 and LiCl exerts protective effects against spatial cognitive deficits induced by icv-STZ in B6 mice. 

### 3.6. Combination of MC5523 and LiCl Prevents the Loss of Cholinergic and Noradrenergic Neurons in icv-STZ B6 Mice

Compared with the mice in the saline/saline/saline control group, the mice in the STZ/saline/saline group showed a significant decrease in the number of cholinergic neurons in the medial septum/diagonal band of Broca (MS/DB) (*p* < 0.001; Figure 5A,B), noradrenergic neurons in the LC (*p* < 0.001; Figure 5A,C), and serotonergic neurons in the Raphe nucleus (*p* < 0.001; Appendix A). Compared with the mice in the STZ/saline/saline group, the mice in the STZ/LiCl/saline group showed a significant reduction in the loss of cholinergic (*p* < 0.001; Figure 5A,B) and serotonergic (*p* < 0.01; Appendix A) neurons but an increase in the loss of noradrenergic neurons (*p* < 0.001; Figure 5A,C). Compared with the STZ/saline/saline-treated mice, the STZ/MC5523/saline-treated mice also showed a significant reduction in the loss of cholinergic (*p* < 0.001; Figure 5A,B), serotonergic (*p* < 0.05; Appendix A), and noradrenergic (*p* < 0.001; Figure 5A,C) neurons. Interestingly, we further found that compared with the icv-STZ mice treated with MC5523 or LiCl alone, the treatment with MC5523 plus LiCl significantly prevented the loss of cholinergic and noradrenergic neurons (*p* < 0.01–0.001; Figure 5A–C). Thus, the prevention of neuronal loss exerted by the cotreatment with MC5523 and LiCl is similar in both icv-STZ OVX 3×Tg-AD and B6 mice.

### 3.7. Combination of MC5523 and LiCl Greatly Increases PSD95 and MAP2 Expression and the NR2A/NR2B Ratio in icv-STZ B6 Mice

To explore the molecular mechanisms underlying the LiCl/MC5523-induced enhancement of neurite outgrowth and synaptic-related markers, the levels of microtubule-associated protein 2 (MAP2), postsynaptic density protein 95 (PSD95), and the N-methyl-D-aspartate receptor 2A/2B (NR2A/NR2B) ratio were measured by western blotting (Figure 5D). The results showed that the PSD95 (*p* < 0.05), MAP2 (*p* < 0.01), and NR2A/NR2B ratio (*p* < 0.01) levels in the STZ/saline/saline group were significantly reduced compared with those in the saline/saline/saline (control) group. The LiCl treatment alone significantly increased the NR2A/NR2B ratio (*p* < 0.01), and the MC5523 treatment significantly increased the MAP2 level and the NR2A/NR2B ratio (*p* < 0.01). However, compared with the mice in the STZ/saline/saline group, the mice in the STZ/LiCl/MC5523 group showed a significant upregulation of the synaptic-related proteins PSD95 (*p* < 0.05) and MAP2 (*p* < 0.001) and an increased NR2A/NR2B ratio (*p* < 0.001). Furthermore, compared with the MC5523 or LiCl treatment alone, the cotreatment with MC5523 and LiCl significantly increased the expression level of MAP2 (*p* < 0.001).

### 3.8. Combination of MC5523 and LiCl Greatly Alleviates Gliosis in icv-STZ B6 Mice

The gliosis induced in the STZ/saline/saline group, which involves astrocytic and microglial activation, was largely decreased in the STZ/LiCl/MC5523 group compared with that in the STZ/LiCl/saline and STZ/MC5523/saline groups (*p* < 0.001; Figure 6A–C). Furthermore, to evaluate the hippocampal inflammation in all groups, the expression levels of the three most common inflammatory factors, i.e., nuclear factor (NF)-κB, interleukin (IL)-6, and tumor necrosis factor (TNF-α), were determined by western blotting (Figure 6D). We found that the levels of IL-6 and TNF-α were significantly increased in the icv-STZ mice compared with those in the saline/saline/saline-treated mice (*p* < 0.05–0.001; Figure 6D). The IL-6 level in the STZ/LiCl/saline (*p* < 0.05; Figure 6D) and STZ/LiCl/MC5523 (*p* < 0.01; Figure 6D) groups was significantly reduced compared with that in the STZ/saline/saline group. In addition, the TNF-α level in the STZ/LiCl/saline, STZ/MC5523/saline, and STZ/LiCl/MC5523 groups was significantly reduced compared with that in the STZ/saline/saline group (*p* < 0.001; Figure 6D). We further found that the TNF-α level in the STZ/LiCl/MC5523 group was significantly reduced compared with that in the STZ/LiCl/saline group (*p* < 0.001; Figure 6D). These data indicate that MC5523 combined with LiCl largely decreased gliosis in both the icv-STZ B6 and icv-STZ OVX 3×Tg-AD mice.

### 3.9. Combination of MC5523 and LiCl Decreases Oligomer Aβ and Tau Phosphorylation Levels by Increasing pS9-GSK3β in icv-STZ B6 Mice

The major pathological features of AD were also assessed by western blotting (Figure 7). We found that the levels of BACE1 (*p* < 0.001; Figure 7A,B) and 6E10 (*p* < 0.01; Figure 7A,B) in the STZ/saline/saline group were significantly increased compared with those in the control group. However, the BACE1 (*p* < 0.001; Figure 7A,B) and 6E10 (*p* < 0.01; Figure 7A,B) levels in the STZ/LiCl/MC5523 group were concurrently decreased. In addition, compared with the mice in the STZ/saline/saline group, the mice in the STZ/LiCl/saline and STZ/MC5523/saline groups also showed significantly reduced BACE1 levels (*p* < 0.001; Figure 7A,B) but not 6E10 levels (Figure 7A,B). These results indicate that the reduction in oligomeric Aβ levels induced by the combination of LiCl and MC5523 involves the Aβ deposition-related enzyme BACE1 and Aβ clearance-related enzymes, such as neprilysin (NEP) and insulin-degrading enzyme (IDE), and other mechanisms could also contribute to the reduced oligomeric Aβ level. However, the levels of NEP and IDE were unchanged in the STZ/LiCl/MC5523 group compared with those in the STZ/saline/saline group (data not shown). In addition, the level of pS9-GSK3β/GSK3β (*p* < 0.05; Figure 7A,C) was significantly decreased in the STZ/saline/saline group, while both the pY216-GSK3β/GSK3β ratio (*p* < 0.05; Figure 7A,C) and tau protein phosphorylation at pS396 and pT181 (*p* < 0.001; Figure 7A,C) were increased compared with those in the control group. In the STZ/LiCl/saline group, the pS9-GSK3β/GSK3β ratio was increased (*p* < 0.01; Figure 7A,C), and the levels of tau protein phosphorylation at pS396 (*p* < 0.001; Figure 7A,C) and pT181 (*p* < 0.05; Figure 7A,C) were decreased compared with those in the STZ/saline/saline group. In addition, in the STZ/MC5523/saline group, the pS9-GSK3β/GSK3β ratio was significantly increased (*p* < 0.001; Figure 7A,C), while the pY216-GSK3β/GSK3β ratio (*p* < 0.01; Figure 7A,C) and tau protein phosphorylation at pS396 (*p* < 0.001; Figure 7A,C) and pT181 (*p* < 0.01; Figure 7A,C) were decreased compared with those in the STZ/saline/saline group. However, in the STZ/LiCl/MC5523 group, the pS9-GSK3β/GSK3β ratio was significantly increased (*p* < 0.001; Figure 7A,C), but the pY216-GSK3β/GSK3β ratio (*p* < 0.001; Figure 7A,C) and tau protein phosphorylation at pS396 (*p* < 0.001; Figure 7A,C), pT231 (*p* < 0.01; Figure 7A,C), and pT181 (*p* < 0.001; Figure 7A,C) were decreased. The administration of LiCl, MC5523, and LiCl with MC5523 had no effect in reducing the CDK5 level, which was increased by the icv-STZ treatment (Figure 7A,C). Furthermore, the cotreatment with MC5523 and LiCl significantly increased the pS9-GSK3β/GSK3β ratio compared with LiCl (*p* < 0.01; Figure 7A,C) or MC5523 (*p* < 0.05; Figure 7A,C) alone. The pT181 level was also significantly reduced in the STZ/LiCl/MC5523 group compared with that in the STZ/LiCl/saline group (*p* < 0.05; Figure 7A,C). Therefore, the cotreatment with LiCl and MC5523 reduced the oligomer Aβ and tau phosphorylation levels potentially by decreasing pS9-GSK3β in the icv-STZ B6 mice.

## 4. Discussion

The aim of this study was to determine whether the administration of MC5523 could yield greater benefits and reduce the adverse effects of LiCl in the treatment of AD. The data revealed that the chronic administration of LiCl (141.3 mg/kg, i.p.) induced greater mortality and hepatotoxicity; however, the treatment with MC5523 reduced the LiCl-induced toxicity in icv-STZ OVX 3×Tg-AD mice. Furthermore, the cotreatment with MC5523 and LiCl greatly reduced the neuronal loss and gliosis in the icv-STZ OVX 3×Tg-AD and B6 mice. Compared with the treatment with LiCl or MC5523 alone, the chronic cotreatment with LiCl and MC5523 improved short-term memory and spatial cognition while reducing the related pathological features of AD in the icv-STZ B6 mice.

In this study, we evaluated the effects of LiCl and/or MC5523 on cognition and the pathological characteristics of AD using two AD mouse models. The intracerebroventricular injection of STZ in wild-type mice is a well-known strategy used to generate a model of sporadic AD [44] and exacerbate disease progression in 3×Tg-AD mice [10]. We found that the icv-STZ treatment in B6 male mice impaired cognition and was associated with decreases in the expression of synaptic-related proteins, including MAP2, PSD95, and NR2A/NR2B, in the hippocampus. The icv-STZ treatment also induced the loss of cholinergic neurons in the MS/DB, noradrenergic neurons in the LC, and serotonergic neurons in the Raphe nucleus. Additionally, icv-STZ increased neuroinflammation, the levels of oligomeric Aβ, activated GSK3β, and CDK5, and tau protein phosphorylation at T181 and S396 in the hippocampus of B6 mice. Previous evidence suggests that icv-STZ treatment changes the gene expression levels of CDK5 and GSK3β in the hippocampus in cynomolgus monkeys [45]. Evidence further suggests that gliosis and postsynaptic neurotoxicity play critical roles in STZ-induced memory impairment and neuronal death [46]. Neuronal loss in the Raphe, LC, and hippocampal CA1 subregion and gliosis were also observed in the icv-STZ 3×Tg-AD mice. The results of this study, including the behavioral, biochemical, and pathological abnormalities in the icv-STZ mice, are consistent with findings reported in previous studies [8,9,44]. Therefore, icv-STZ can be used in both sporadic (wild-type B6 mice) and familial (OVX 3×Tg mice) AD animal models to evaluate the effects of LiCl and/or MC5523.

We found that the chronic administration of LiCl (141.3 mg/kg, i.p.) improved only short-term memory and was associated with increased protection of cholinergic and serotonergic neurons and reduced levels of gliosis, BACE1, and tau protein phosphorylation in icv-STZ B6 mice. These results are consistent with previous findings showing that LiCl treatment completely blocked the increase in BACE1 expression induced by traumatic brain injury [47]. Evidence has also shown that LiCl plays a prominent role in modulating inflammation in neurodegenerative and mood disorders [48]. Furthermore, a previous study has shown that LiCl mediates the reductions in GSK3β, tau phosphorylation, and cell death in neurodegenerative diseases [49]. However, we found that the chronic administration of LiCl induced more noradrenergic neuronal loss in the icv-STZ B6 mice and more serotonergic neuronal loss, higher mortality and hepatotoxicity in the icv-STZ OVX 3×Tg-AD mice. One study also found that a higher dose of LiCl (e.g., 120 mg/kg, i.p.) was toxic to mice [50]. In addition, while the administration of LiCl induced beneficial effects against ischemia/reperfusion injury in the central nervous system, heart, and kidneys, it increased apoptosis and oxidative stress in the liver [51]. Therefore, side effects limit the therapeutic applications of LiCl. 

Regarding the in vitro results, we speculate that the different activity of MCs could be attributed to the levels of the main active content in the fruits. This phenomenon commonly occurs in natural products, such as Crocus [52] and tomato [53]. Our in vivo results show that MC5523 and LiCl alone have marginal effects on cognitive deficits and pathological characteristics in icv-STZ B6 mice. In addition, the administration of MC5523 alone attenuated gliosis and neuronal loss in the icv-STZ mice. MC has potent neuroprotective activity against neurological deficits in diabetic mice [32]. Evidence also suggests that MC can improve obesity-associated peripheral inflammation and neuroinflammation [54]. Thus, MC5523 and LiCl alone have similar effects on short-term memory, gliosis, and neuronal loss in icv-STZ mice. However, hepatotoxicity was induced only by LiCl alone but not by MC5523.

Based on the results of the LiCl plus MC5523 treatment, we found that the administration of MC5523 increased the survival rate and reduced the hepatotoxicity induced by LiCl in the icv-STZ OVX 3×Tg-AD mice. A previous study showed that ucche (*Momordica charantia* L. var. muricata (Willd.) Chakravarty) supplementation had a protective effect against CCl_4_-induced hepatotoxicity [55]. Evidence also suggests that ethanol extracts of Hualien Nos. 3 and 4 have hepatoprotective effects against alcoholic fatty liver disease [34]. The great beneficial effects of LiCl combined with MC5523 observed in the icv-STZ B6 mice included improved short-term memory and spatial cognition; reduced levels of Aβ oligomer, tau protein phosphorylation at Thr-181, and gliosis; and increased expression of synaptic plasticity-related proteins, such as MAP2 and PSD95, and the inactive form of GSK3β. BACE1, which is the main enzyme involved in Aβ generation, is decreased in AD patients [56]. In this study, we found that the levels of pS9-GSK3β and the amount of Aβ oligomers were concurrently reduced by the administration of LiCl plus MC5523 but not LiCl or MC5523 alone. However, the treatment with LiCl, MC5523, and MC5523 plus LiCl had no effect on Aβ-degrading enzymes, such as IDE and NEP, in the hippocampus of the icv-STZ B6 mice (data not shown). Furthermore, GSK3β has been shown to participate in the transcriptional regulation of BACE1, leading to enhanced amyloidogenic processing [57,58]. Thus, GSK3β may play a role in mediating the reduction in the amount of Aβ oligomers following cotreatment with LiCl and MC5523. In addition, we found that compared with either treatment alone, the cotreatment with LiCl and MC5523 largely increased the expression levels of PSD95 and MAP2 in the hippocampus of the icv-STZ B6 mice. However, the expression of presynaptic proteins, such as synaptophysin, was not changed by the treatment with LiCl and MC5523 either alone or in combination in the icv-STZ mice (data not shown). Recent evidence has shown that MAP2 and PSD95 play important roles in synaptic plasticity in AD mice [59]. Furthermore, the expression levels of CDK5 were increased in the icv-STZ mice. However, the treatment with LiCl, MC5523, or LiCl with MC5523 could not decrease the CDK5 expression level increased by the icv-STZ treatment. The expression of CDK5, which is a serine/threonine kinase, has been demonstrated to play an important role in multiple functions, including neural development, neurodegeneration, learning, memory, adult synaptic plasticity, and neurotransmitter release in the adult synapse [60,61,62,63,64]. Thus, the beneficial effects of LiCl combined with MC5523 were CDK5-independent in the icv-STZ B6 mice. Furthermore, the tau protein phosphorylation at Thr-181 induced by icv-STZ after the treatment with LiCl and MC5523 was much lower than that following the treatment with LiCl or MC5523 alone. Evidence shows that abnormal tau protein phosphorylation at Thr-181 plays an important role in microtubule dysfunction [65]. The superior neuroprotective effects are consistent with those reportedly induced by combining LiCl with other histone deacetylase inhibitors or L-dopa [50,66]. Many lines of evidence further indicate that combination therapies with multiple drug targets have better therapeutic outcomes in the treatment of AD [67,68,69,70]. However, the safety of food supplements must be considered in foods contaminated with Ochratoxin A, pesticides or heavy metals [71,72]. Therefore, studies should address this critical issue and contemplate how to safeguard the population.

## 5. Conclusions

In conclusions, the combined treatment with LiCl and MC5523 is a rational strategy for obtaining robust beneficial effects in terms of survival rate, neuroprotection, and gliosis in AD. Taken together, these results show that the combination of LiCl and MC5523 has potential as a more effective therapeutic strategy for the treatment of patients with AD.

## Figures and Tables

**Figure 1 nutrients-10-01888-f001:**
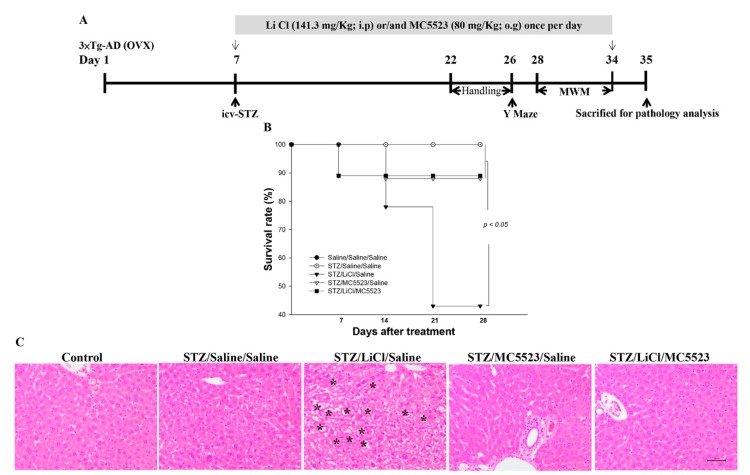
MC5523 increased the survival rate and reduced hepatotoxicity in LiCl-treated icv-STZ OVX 3×Tg-AD mice. (**A**) The timeline of the treatment with LiCl, MC5523, and LiCl plus MC5523 in the icv-STZ OVX 3×Tg-AD mice. (**B**) Kaplan-Meier graph showing the survival pattern in each group of icv-STZ OVX 3×Tg-AD mice treated with LiCl, MC5523, and LiCl plus MC5523. (**C**) The results of staining the mouse liver tissue with hematoxylin and eosin (H&E). The results reveal that MC5523 reduced the hepatotoxicity induced by LiCl. * indicates a hepatotoxic signal in the liver.

**Figure 2 nutrients-10-01888-f002:**
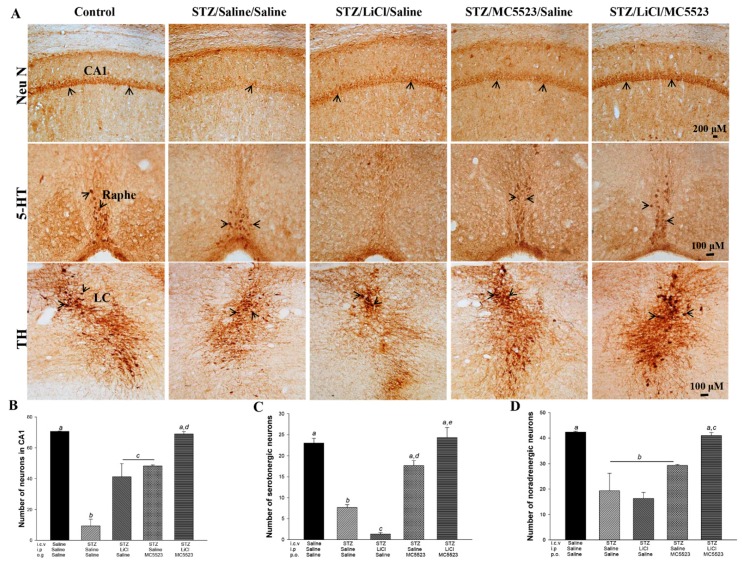
Cotreatment with LiCl and MC5523 greatly prevented neuronal loss in the icv-STZ OVX 3×Tg-AD mice. (**A**) Representative images of pyramidal neurons in the CA1 subregion of the hippocampus, serotonergic neurons in the Raphe nucleus, and noradrenergic neurons in the locus coeruleus (LC). Scale bars are 200 μm in the CA1 subregion and 100 μm in the Raphe nucleus and LC. (**B**–**D**) The number of pyramidal neurons in the CA1 subregion of the hippocampus, serotonergic neurons in the Raphe nucleus, and noradrenergic neurons in the LC. The results reveal the robust beneficial effect of LiCl combined with MC5523 on pyramidal, serotonergic, and noradrenergic neurons (*n* = 3 per group). Arrowheads indicate positive staining signals. The quantitative data are shown as the mean ± SEM (*n* = 3 per group). Means that do not share a letter are significantly different (*p* < 0.05).

**Figure 3 nutrients-10-01888-f003:**
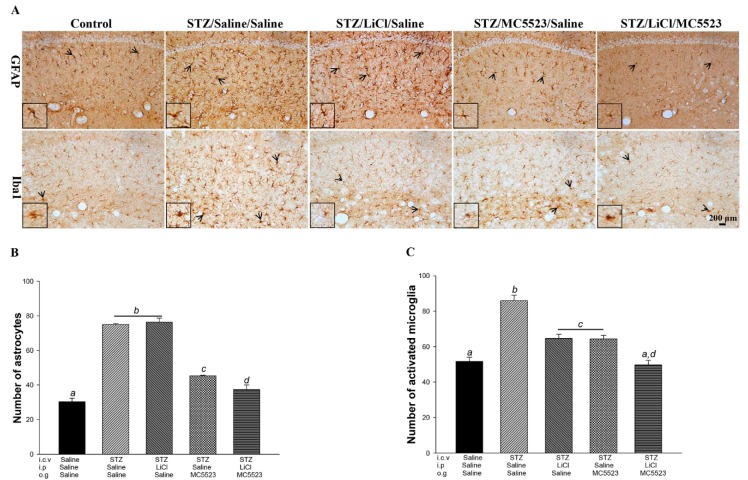
MC5523 greatly decreased the level of gliosis in LiCl-treated icv-STZ OVX 3×Tg-AD mice. (**A**) Representative images of astrocytes (with glial fibrillary acidic protein (GFAP) staining) and microglia (with Iba1 staining) in the hippocampus. Scale bar = 200 μm. Arrowheads indicate positive staining signals. (**B**–**C**) The numbers of activated astrocytes and microglia in the hippocampus. The results reveal the beneficial effect of LiCl combined with MC5523 on reducing neuroinflammation. The quantitative data are shown as the mean ± standard error of the mean (SEM) (*n* = 3 per group). Means that do not share a letter are significantly different (*p* < 0.05).

**Figure 4 nutrients-10-01888-f004:**
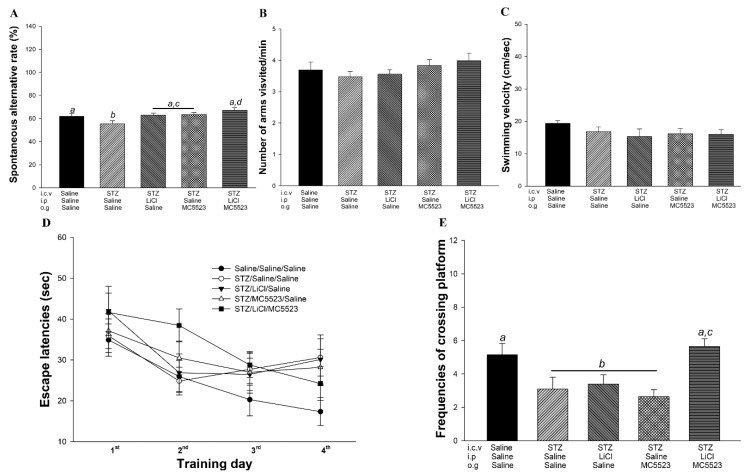
MC5523 greatly improved short-term memory and spatial cognition in LiCl-treated icv-STZ B6 mice. (**A**) Spontaneous alternation rate in icv-STZ B6 mice treated with MC5523 and LiCl. Short-term memory retrieval in the groups treated with LiCl and MC5523 was better than that in the groups treated with LiCl or MC5523 alone. (**B**) The number of arms visited by icv-STZ mice treated with LiCl and/or MC5523. There were no specific preferences for any arms among the mice. (**C**) Swimming velocity of the mice in the Morris water maze (MWM). The mice in all groups showed the same swimming ability. (**D**) Learning trends of the mice over the 4 MWM training days. The escape latencies were decreased in both the STZ/Li/MC5523 and saline/saline/saline groups from training days 1 to 4. (**E**) Long-term memory retrieval results. A probe trial was conducted 48 h after the final training trial to evaluate long-term memory retrieval. Long-term memory retrieval impairment was observed in the STZ/saline/saline group, and the treatment with LiCl combined with MC5523 attenuated the deficit. The data are shown as the mean ± SEM (*n* = 12–15 per group). Means that do not share a letter are significantly different (*p* < 0.05).

**Figure 5 nutrients-10-01888-f005:**
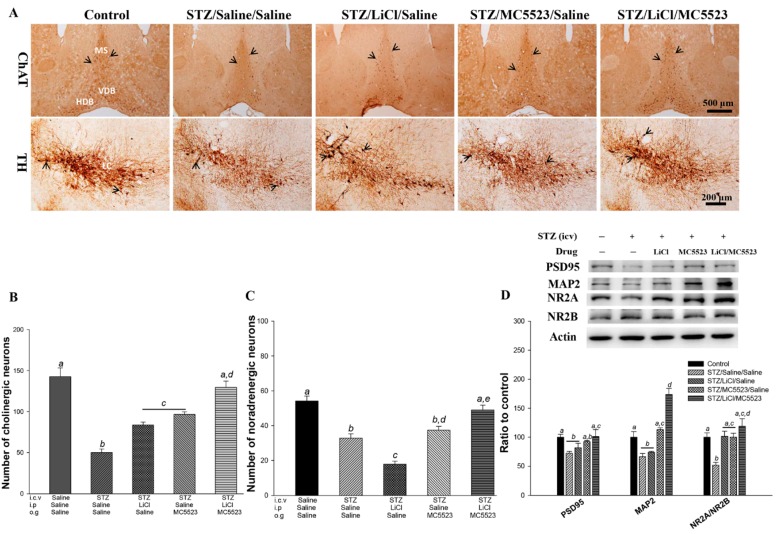
MC5523 prevented neuronal loss and increased the MAP2 level in the LiCl-treated icv-STZ B6 mice. (**A**) Representative images of cholinergic neurons (with ChAT staining) in the MS/DB and noradrenergic neurons (with TH staining) in the locus coeruleus (LC). Scale bars are 500 μm for ChAT and 200 μm for TH, and the arrowheads indicate positive staining signals. (**B**–**C**) Number of cholinergic and noradrenergic neurons. The results show the beneficial effect of LiCl combined with MC5523 on cholinergic and noradrenergic neurons. (**D**) Representative western blots and densitometry results of PSD95 and MAP2 and the NR2A/NR2B ratio with β-actin serving as an internal control. The quantitative data are shown as the mean ± SEM (*n* = 3–5 per group). Means that do not share a letter are significantly different (*p* < 0.05).

**Figure 6 nutrients-10-01888-f006:**
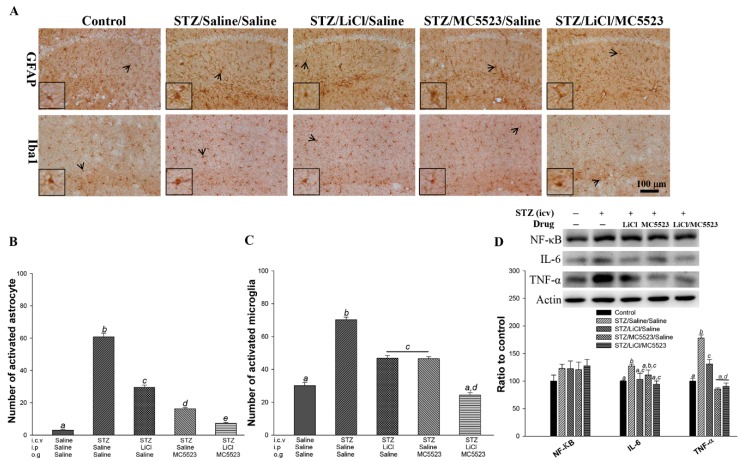
MC5523 greatly decreased the level of gliosis in the LiCl-treated icv-STZ B6 mice. (**A**) Representative images of astrocytes (with GFAP staining) and microglia (with Iba1 staining) in the hippocampus. Scale bar = 100 μm, and the arrowheads indicate positive staining signals. (**B**–**C**) The number of activated astrocytes and microglia in the hippocampus. (**D**) Representative western blots and densitometry results of NF-κB, IL-6, and TNF-α with β-actin serving as an internal control. The quantitative data are shown as the mean ± SEM (*n* = 3–5 per group). Means that do not share a letter are significantly different (*p* < 0.05).

**Figure 7 nutrients-10-01888-f007:**
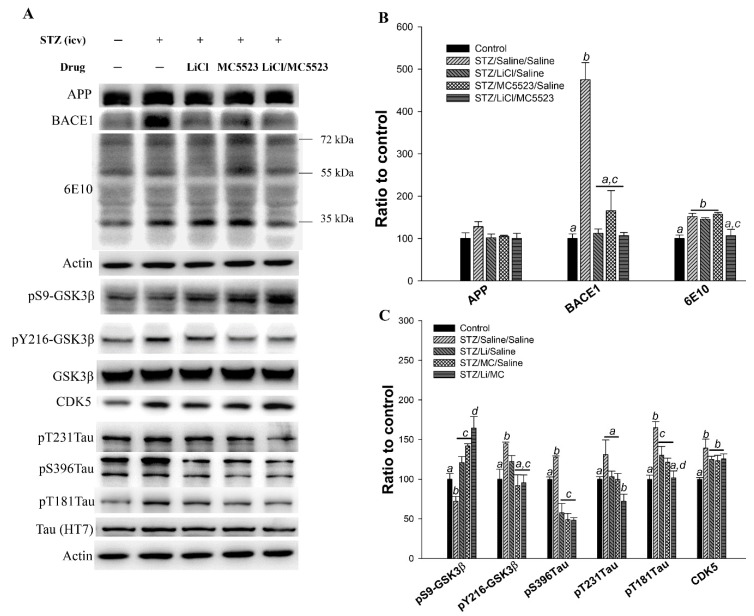
LiCl combined with MC5523 decreased the pathological features of AD in icv-STZ B6 mice. Representative western blots (**A**) and densitometry results (**B**–**C**) of APP, BACE1, 6E10, pS9-GSK3β, pY216-GSK3β, CDK5, and HT7 and pTau at S396, T231, and T181 with β-actin serving as an internal control. The quantitative data are shown as the mean ± SEM (*n* = 3–5 per group). Means that do not share a letter are significantly different (*p* < 0.05).

**Table 1 nutrients-10-01888-t001:** List of primary antibodies.

Antibody	Species	Supplier	WB Dilution	IHC Dilution	IF Dilution
APP	Rabbit	Sigma-Aldrich	1:1000	－	
BACE1	Rabbit	Cell Signaling	1:1000	－	
6E10	Mouse	COVANCE	1:1000		
5-HT	Rat	Millipore	－	1:200	
ChAT	Rabbit	Millipore	－	1:1000	
TH	Rabbit	Millipore	－	1:1000	
NF-κB	Rabbit	Cell Signaling	1:1000	－	
IL-6	Goat	Santa Cruz	1:1000		
TNF-α	Goat	Santa Cruz	1:1000		
IL-1β	Rabbit	Santa Cruz	1:1000		
MnSOD	Rabbit	Millipore	1:1000	－	
GFAP	Mouse	Millipore	－	1:1000	
Iba-1	Rabbit	Wako	－	1:1000	
CDK5	Mouse	Millipore	1:1000	－	
pS9-GSK3β	Rabbit	Cell Signaling	1:1000	－	
GSK3β	Rabbit	Cell Signaling	1:1000	－	
PY216-GSK3β	Mouse	Millipore	1:1,000		
pT181Tau	Rabbit	Millipore	1:1000	－	
pS396Tau	Rabbit	Invitrogen	1:1000	－	
pT231Tau	Rabbit	Invitrogen	1:1000	－	
Total tau (HT7)	Mouse	Thermo	1:500		
NR2A	Rabbit	Millipore	1:1000		
NR2B	Rabbit	Millipore	1:1000	－	
PSD95	Goat	Santa Cruz	1:1000	－	
MAP2	Rabbit	Millipore	1:1000	－	1:1000
Neu N	Mouse	Millipore	1:1000		1:1000
β-Actin	Mouse	Millipore	1:2000	－	

WB, western blot; IHC, immunohistochemistry; IF, immunofluorescence.

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
