# Peer review of "Administration of Momordica charantia Enhances the Neuroprotection and Reduces the Side Effects of LiCl in the Treatment of Alzheimer’s Disease"

_nutrients, 2018, doi:10.3390/nu10121888_

Round 1

Reviewer 1 Report

In this manuscript, the authors investigated the in vitro and in vivo pharmacological functions of LiCl combined with Momordica charantia (MC) in treating AD. The in vitro results showed that MC5523 under hyperglycemia or tau hyperphosphorylation. The combined treatment with LiCl and MC5523 is a rational strategy for obtaining robust beneficial effects in terms of survival rate, neuroprotection, and neuroinflammation in the AD. Taken together, the results show that the combination of LiCl and MC5523 has potential as a more effective therapeutic strategy for patients with the AD. This research is well designed and positive results were obtained. I would like to recommend minor revisions.

There are still several issues need to be addressed.

1. In the animal model, the dosage of LiCl was 141.3 mg/kg (i.p., daily) (Figure 1).    Why the dosage was confirmed as 141.3 mg/kg? Why it was not 80 or 100, but just 141.3 mg/kg? 2. Could the authors try to infer possible reason(s) why different MC might display different activity? If the contents of the main active natural products in MC2, 3, 5 and 5523 are different?

Reviewer 2 Report

The manuscript "Administration of Momordica Charantia Enhances the Neuroprotection and Reduces the Side Effects of LiCl in Treating Alzheimer’s Disease" by Huang and collaborators describes a well-designed study about the importants of natural food supplementation to prevent toxic effects of drugs administration.

I find the article suitable for Nutrients, but it needs several important revisions.

In particular my comments are:

1)  Lines 14-28: the abstract is not useful for the reader. The authors did not report any result and the information about the experimental design are poor.

2) Lines 33-39: the information about AD and therapy are insufficient. The authors have to extend this section! No info is reported about the different theories of AD onset and progression (eg: cholinergic, amyloid, metal hypotheses) and the consequent symptomatic drugs used in therapy (AChE inhibitors or NMDAr blockers). In my opinion they could introduce in this section something about the innovative research about "Type 3 diabetes" and AD. It can be coherent with the aim of their research (eg: they studied the inflammatory mediators and CDK5).

3) Lines 40-43: The authors should report recent researches about small molecules with GSK3β as AD target.

4) Figure 2D, 3A/B/C etc.: The statistics are not homogenous. The authors should report the same letter when the data are not significantly different.

5) Line 274: The authors introduce the F(x,x) parameter without explain what it is.

6) Line 340-344/Figure 6D: The authors should speculate in a correct way, including the significance in the statistical analyses for each different experiment. Like in several data reported in the figure 7 there is not a synergistic effect in LiCl or MC5523 treatments. In numerous experiments the effects seem due only to one of the treatments.

7) Figure 7: See point 6. In numerous experiments the effects seem due only to one of the treatments.

8) Line 456-457: “In this study, we found that the levels of BACE1 and Aβ deposition were concurrently reduced by the administration of LiCl plus MC5523 but not LiCl or MC5523 alone”. I did not understand this speculation. It is not coherent with the data reported in Figure 7. Moreover, there are not experiment reported about Abeta deposition.

9) Line 480: “Many lines of evidence further indicate that combination therapies with multiple drug targets have better therapeutic outcomes in treating AD”. It is correct, but the references are not suitable. I suggest searching for a couple of reviews about AD and multi-target therapy.

10) Discussion: The authors should describe the “other hand” in the use of food supplements in the therapy of neurodegenerative/chronic diseases. I suggest searching for paper/reviews about the problem of the quality of these products, in particular the contamination with mycotoxins (such as Ochratoxin A), pesticides and heavy metals. There are several papers in literature about this topic published in the recent past.

11) References: There are several typos (eg ref 55 and 56).
